# Baseline mean platelet volume is a strong predictor of major and life-threatening bleedings after transcatheter aortic valve replacement

Antonin Trimaille[1,2], Kensuke Matsushita[1,2], Benjamin Marchandot[1], Adrien Carmona[1], Sébastien Hess[1], Marion Kibler[1], Joé Heger[1], Antje Reydel[1], Laurent Sattler[3], Lelia Grunebaum[3], Laurence Jesel[1,2], Patrick Ohlmann[1], Olivier Morel[1,2]*

1 Department of Cardiovascular Medicine, Nouvel Hôpital Civil, Strasbourg University Hospital, Strasbourg, France, 2 INSERM (French National Institute of Health and Medical Research), UMR 1260, Regenerative Nanomedicine, FMTS, Strasbourg, France, 3 Department of Hemostasis, Hôpital de Hautepierre, Strasbourg University Hospital, Strasbourg, France

* olivier.morel@chru-strasbourg.fr

## Abstract

### Background

Bleeding following transcatheter aortic valve replacement (TAVR) has important prognostic implications. This study sought to evaluate the impact of baseline mean platelet volume (MPV) on bleeding events after TAVR.

### Methods and results

Patients undergoing TAVR between February 2010 and May 2019 were included. Low MPV (L-MPV) was defined as MPV $\leq$ 10 fL and high MPV (H-MPV) as MPV >10 fL. The primary endpoint was the occurrence of major/life-threatening bleeding complications (MLBCs) at one-year follow-up. Among 1,111 patients, 398 (35.8%) had L-MPV and 713 (64.2%) had H-MPV. The rate of MLBCs at 1 year was higher in L-MPV patients compared with H-MPV patients (22.9% vs. 17.7% respectively, p = 0.034). L-MPV was associated with vascular access-site complications (36.2% vs. 28.9%, p = 0.012), early (<30 days) major bleeding (15.6% vs. 9.4%, p<0.01) and red blood cell transfusion >2 units (23.9% vs. 17.5%, p = 0.01). No impact of baseline MPV on overall death, cardiovascular death and ischemic events (myocardial infarction and stroke) was evidenced. Multivariate analysis using Fine and Gray model identified preprocedural hemoglobin (sHR 0.84, 95%CI [0.75–0.93], p = 0.001), pre-procedural L-MPV (sHR 1.64, 95%CI [1.16–2.32], p = 0.005) and closure time adenosine diphosphate post-TAVR (sHR 2.71, 95%CI [1.87–3.95], p<0.001) as predictors of MLBCs.

### Conclusions

Preprocedural MPV was identified as an independent predictor of MLBCs one year after TAVR, regardless of the extent of platelet inhibition and primary hemostasis disorders.

Furthermore, currently, we do not have authorization to share any personal data with third external parties as the French legislation (Jardé law) does not allow a free sharing of human research participant data. Patients could be potentially identified based their age, sex, type of outcomes etc. in a single center study. Data could be made available upon request by submitting an email to President of the Scientific Board - Committee for Scientific Research of Hôpitaux Universitaires de Strasbourg, 1, place de l'hôpital, BP 426, 67091 Strasbourg cedex. Email: emmanuel.andres@chrustrasbourg.fr."

**Funding:** The authors received no specific funding for this work.

**Competing interests:** The authors have declared that no competing interests exist.

**Abbreviations:** AS, aortic stenosis; CKD, chronic kidney disease; CT-ADP, closure time adenosine diphosphate; HMW, high molecular weight; MLBCs, major and life-threatening bleeding complications; MPV, mean platelet volume; PRI, platelet reactivity index; TAVR, transcatheter aortic valve replacement; VARC-2, Valve Academic Research Consortium-2; vWF, von Willebrand factor.

# Introduction

Bleeding events after transcatheter aortic valve replacement (TAVR) still occur in a sizeable proportion of patients and mitigate long-term benefits [1,2] despite a substantial reduction of vascular complications over the last decade. Several reports have demonstrated that major bleedings following TAVR outweighed ischemic complications and dramatically impacted long-term prognosis [3]. As such, bleeds have been identified as an important area for improved patient care. The optimal antithrombotic regimen after TAVR is not clearly defined. Thrombotic and bleeding risk assessments in TAVR patients are complex and individually tailored antithrombotic therapy according to a specific risk profile are mandatory.

In the search of accurate biomarkers of bleeding risk after TAVR, several studies have investigated the impact of platelets. Thrombocytopenia occurred in 40 to 70% of patients undergoing TAVR [4,5] and is significantly associated with increased mortality [5], thrombosis events such as subclinical leaflet thrombosis [6] and stroke [7], and bleeding [8]. Moreover, it has been previously showed that platelets produced after thrombocytopenia were more reactive [9].

Mean platelet volume (MPV) has been proposed as a surrogate marker of platelet reactivity [10]. It is a machine-calculated measurement of the average platelet size that ranges in normal condition between 7 and 11 fL [11]. Owing to their enrichment in granules, vasoactive and prothrombotic factors, large platelets have been advocated as promising prothrombotic markers in various cardiovascular diseases and metabolic disorders [10]. Consistent with this paradigm, high MPV was associated with shortened bleeding time [9]. As previously demonstrated, increased shear stress during aortic stenosis (AS) is associated with a loss of the high molecular weight (HMW) of the von Willebrand factor (vWF) due to proteolysis of vWF multimers by the metalloproteinase ADAMTS13 [12]. Because large vWF multimers are stored in alpha-granules, low MPV may be the consequence of their release following flow disturbances. Conflicting evidence remain regarding the impact of MPV on bleeding and thrombotic risk in TAVR. Whilst an independent and paradoxical association between baseline high MPV and early bleeding after TAVR was evidenced in a limited cohort of patients [13], a larger trial later depicted a noxious impact of low MPV on periprocedural major and life-threatening bleeding complications (MLBCs) [14]. In the absence of systematic characterization of P2Y12 inhibition by clopidogrel or the assessment of vWF proteolysis, the independent contribution of MPV to bleeding events remains difficult to ascertain.

In the present study, we aimed to assess the independent predictive value of baseline MPV on bleeding complications after TAVR, and to investigate determinants of baseline MPV.

# Methods

## Study population

All consecutive patients with severe aortic stenosis and high or intermediate surgical risk according to Logistic EuroSCORE who underwent TAVR between February 2010 and May 2019 at our institution (Nouvel Hôpital Civil, Strasbourg University, France) were enrolled in a prospective registry. All participants gave their informed written consent before the procedure and agreed to the anonymous processing of their data in the FRANCE-2 registry. The study was approved by the CNIL's committee (French National Commission on Informatics and Liberty) (number 911262). The study protocol conforms to the ethical guidelines of the 1975 Declaration of Helsinki.

All patients received Aspirin (75–160 mg) and Clopidogrel (loading dose 300 mg, 75 mg/day maintenance dose) before the procedure, with ongoing double antiplatelet therapy after

the procedure for 3 months. According to the local protocol, only Aspirin was administered in patients with oral anticoagulant.

Commercially available valves, such as the balloon expandable Edwards SAPIEN XT or S3 prosthesis (Edwards Life sciences LLC, Irvine, CA), the self-expandable CoreValve Evolute-R or Evolut Pro (Medtronic CV, Minneapolis, MN) and the self-expandable supra-annular ACURATE neo valve (Boston Scientific, Marlborough, MA) were used. During the intervention, 100 international units (UI)/kg of unfractionned heparin were administered to achieve an activated clotting time of 250 to 350 seconds. At the end of the procedure, heparin was antagonized with protamine (100 UI/kg). All procedures were performed under analgesic sedation with remifentanil chlorhydrate (0.10 to 0.15 μg/kg/min).

## Collection of data

Baseline clinical and biological data, cardiovascular risk factors, co-morbidities and follow-up variables were recorded and entered into a secure, ethics-approved database. Clinical endpoints including mortality, stroke, bleedings and access-related complications were assessed according to the definitions provided by the Valve Academic Research Consortium-2 (VARC-2) [15]. Bleeding complications were classified as follows: life-threatening bleeding, major bleeding and minor bleeding. We added to the VARC-2 classification the occurrence of occult bleeding requiring more than two red blood cells (RBCs) units. Clinical events were adjudicated by an events validation committee. All patients were contacted by phone to complete a standardized survey about their health status, symptoms, medications and the occurrence of adverse events. In case of no response, data were obtained from primary care physicians or hospital records.

## Study objective and endpoints

The primary objective was to evaluate the association between baseline MPV and bleeding events after TAVR.

The primary endpoint of the study was the occurrence of MLBCs at one-year follow-up.

The secondary endpoints were all-cause death, cardiovascular death (defined as death resulting from myocardial infarction, sudden cardiac death, heart failure, stroke or resulting from other cardiovascular causes), stroke and myocardial infarction during the first year after the procedure.

## Blood samples and biological measurements

Whole blood samples were collected by venipuncture the day before TAVR and immediately processed. Several biological tests were performed including hemoglobin level, platelet parameters (platelet count (PC) and MPV), prothrombin time (PT), activated partial thromboplastin time (aPTT), C-reactive protein (CRP), white blood cell count (WBC), closure-time adenosine diphosphate (CT-ADP). CT-ADP was measured both the day before and the day after TAVR. Analysis of CT-ADP and CT-epinephrine with the primary haemostasis point-of-care assay PFA-100 (Siemens Healthcare, Marburg, Germany) were performed. We have recently described that a threshold value for CT-ADP of more than 180 s could identify patients at higher risk of early and late bleeding events [12,16]. In addition, the extent of P2Y12 inhibition by clopidogrel was evaluated by the analysis of VASP phosphorylation by flow cytometry as previously described [17]. In TAVR, we have identified low platelet reactivity index (PRI) value as a potent indicator of periprocedural complications [17].

According to previous studies on MPV in different cardiovascular diseases reviewed by Chu et al. [10], the common threshold value used to differentiate low and high MPV is 10 fL.

Thus, we divided the study population in two groups according to baseline MPV: low MPV (L-MPV) was defined as MPV ≤10 fL and high MPV (H-MPV) as MPV >10 fL.

## Statistical analysis

Continuous variables were expressed as mean ± standard deviation (SD). Categorical variables are expressed as absolute values and percentages. Continuous variables were compared with the use of parametric unpaired Student's t-test or non-parametric Mann-Whitney tests as appropriate. Categorical variables were compared with chi-square test or Fischer's exact test. Time to event was defined as the time from TAVR to the date of event, with patients censored at death, loss to follow-up or end of study. Kaplan-Meier analyses were used to construct survival plots of time to MLBCs after TAVR and comparison used log-rank test. Association of potential prognostic factors (main demographic, procedural and post- procedural characteristics) and the risk of MLBCs were investigated using Fine and Gray's sub-distribution hazard models. All-cause death was considered as a competing risk. Variables with a p-value <0.05 in the univariate analysis were included in the multivariable model. Results are presented as sub-distribution hazards ratios (sHR) with their 95% confidence intervals (CI). Additional sensitivity analyses for MPV ≤10 fL were performed with DAPT and anticoagulant therapy, which are known to impact MLBCs. Propensity score matching was performed to adjust DAPT and anticoagulant therapy between patients with and without MPV ≤10 fL. The propensity score model was developed using logistic regression and the model included DAPT and anticoagulant therapy as covariates. A nearest-neighbor algorithm was used to match patients with and without MPV ≤10 fL in a 1:1 ratio, with a caliper width equal to 0.2 of the SD of the logit of the propensity score. Since reduced vascular complications were described with new generation devices (Sapien 3, Evolut-Pro and Acurate-Neo), additional subanalyses were performed on patients treated with these devices. A p-value <0.05 was considered significant. Statistical analyses were performed using JMP 13 software (SAS Institute, Cary, NC), GraphPad Prism (version 6.0 for Mac, GraphPad Software, La Jolla California USA), and R software version 3.6.3.

## Results

### Baseline characteristics

Of the initial 1,125 patients enrolled in the study, baseline MPV was available in 1,111 patients: 398 (35.8%) patients were qualified as L-MPV (MPV ≤10 fL) and 713 (64.2%) as H-MPV (MPV >10 fL) (Fig 1). Follow-up was completed at one year for 1,078 patients (97.0%).

Both groups were well balanced for baseline characteristics expect for chronic kidney disease (CKD) that was less frequent in L-MPV patients (15.8% vs 21.2%, p = 0.03) (Table 1). There was no difference between the two groups for preprocedural left ventricular ejection fraction (LVEF) and routinely used echo parameters aimed to assess AS severity including mean gradient and aortic valve area. Left ventricular (LV) mass, left ventricular end-diastolic diameter (LVEDD) and end-systolic diameter (LVESD) were lower in L-MPV patients (p<0.05 for all). Prehospital antithrombotic management and procedural characteristics were comparable among subgroups (S1 Table). At discharge, L-MPV patients were less frequently treated with anticoagulant therapy (41.5% vs. 48.0%, p = 0.041) and more frequently with clopidogrel (57.8% vs. 51.5%, p = 0.031). Baseline biological parameters in L-MPV patients showed lower hemoglobin (11.9 ± 1.7 g/dL vs. 12.1 ± 1.7 g/dL, p = 0.016) and BNP (532 ± 780 pg/mL vs. 753 ± 996 pg/mL, p<0.001) levels but increased platelets count (249 ± 78 G/L vs. 215 ± 69 G/L, p<0.001) (Table 2).

## Impact of baseline MPV on outcomes after TAVR

The rate of MLBCs at 1-year follow-up was 22.9% in L-MPV patients as compared with 17.7% in H-MPV patients (p = 0.034) (Table 3, Fig 2). Additionally, L-MPV was significantly associated with a higher risk of vascular access-site complications (36.2% vs. 28.9%, p = 0.012), early major bleeding events (15.6% vs. 9.4%, p<0.01) and red blood cell transfusion of more than two units (23.9% vs. 17.5%, p = 0.01) (Table 3). By contrast, no impact of baseline MPV on death from any cause, cardiovascular death and ischemic events (myocardial infarction and stroke) could be evidenced (p>0.05 for all).

Multivariate analysis identified logistic EuroSCORE (sHR 1.01, 95% CI [1.00–1.03], p = 0.03), preprocedural hemoglobin (sHR 0.84, 95% CI [0.75–0.93], p = 0.001), preprocedural L-MPV (sHR 1.64, 95% CI [1.16–2.32], p = 0.005) and CT-ADP post-TAVR >180 sec (sHR 2.71, 95% CI [1.87–3.95], p<0.001) as predictors of MLBCs (Table 4). When the cohort was split into four subgroups according to baseline MPV (threshold 10 fL) and post-TAVR CT-ADP (threshold 180 s), the bleeding rate was at its highest level in patients with baseline MPV <10 fL and post-TAVR CT-ADP >180 s (Fig 3).

Results of our sensitivity analyses are shown in S2 and S3 Tables. After propensity score matching, 389 patients with L-MPV were compared with 389 patients with H-MPV. The magnitude of preprocedural L-MPV was confirmed with additional sensitivity analyses adjusting for DAPT and anticoagulant therapy (sHR: 1.34; 95% CI [0.98–1.84]; p = 0.07) (S3 Table) but was not statistically significant because of the smaller sample size.

Finally, we performed a subanalysis on the 618 patients treated with new generation valves (Sapien 3, Evolut-Pro and Accurate-Neo). Among them, 217 (35.1%) patients had L-MPV at baseline and 401 (64.9%) had H-MPV. While major bleeding <30 days was significantly lower in H-MPV patients in comparison with L-MPV patients (9.0% vs 15.7%, p = 0.022), MLBCs at one-year follow-up did not reach significance difference (17.2% vs 22.1%, p = 0.27) probably due to the limited size of this sub-group (S4 Table).

## Predictors of low mean platelet volume at baseline

By univariate analysis, CKD history, hemoglobin, platelets count and BNP at baseline were significantly linked to L-MPV at baseline (Table 5). No impact of aortic stenosis severity and

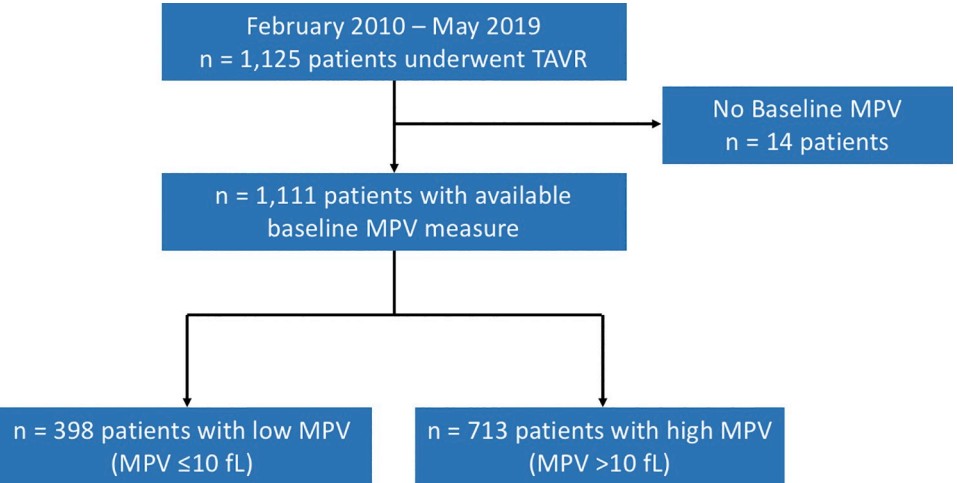

**Fig 1. Flowchart of the study.** This study included 1,111 patients with available baseline MPV measurement and successful TAVR procedure between February 2010 and May 2019. MPV was ≤10 fL (defined as low MPV) in 398 patients and >10 fL (high MPV) in 713 patients. Follow-up was completed for the primary outcome at one year for 1,078 patients (97.0%). *Abbreviations*: MPV = mean platelet volume; TAVR = transcatheter aortic valve replacement.

**Table 1. Demographic characteristics.**

| Variables | Baseline MPV | | p value |
|---|---|---|---|
| | ≤10 fL L-MPV (n = 398) | >10 fL H-MPV (n = 713) | |
| **Demographic characteristics** | | | |
| Age–years | 82.6 ± 7.2 | 83.0 ± 7.0 | 0.48 |
| Male sex–n (%) | 181 (45.5) | 326 (45.7) | 0.94 |
| BMI–kg/m$^2$ | 26.5 ± 5.3 | 27.1 ± 5.9 | 0.08 |
| Logistic EuroSCORE–% | 18.4 ± 13.2 | 20.0 ± 14.1 | 0.053 |
| EuroSCORE 2 –% | 5.6 ± 7.3 | 6.0 ± 6.5 | 0.36 |
| STS mortality–% | 6.5 ± 6.7 | 6.5 ± 5.3 | 0.93 |
| **Cardiovascular risk factors** | | | |
| Current smoking–n (%) | 14 (3.5) | 27 (3.8) | 0.82 |
| Hypertension–n (%) | 323 (81.1) | 595 (83.5) | 0.33 |
| Dyslipidemia–n (%) | 232 (58.5) | 427 (59.9) | 0.60 |
| Diabetes mellitus–n (%) | 131 (32.9) | 219 (30.7) | 0.45 |
| **Medical history** | | | |
| Coronary artery disease–n (%) | 201 (50.5) | 331 (46.4) | 0.67 |
| Peripheral artery disease–n (%) | 108 (27.1) | 187 (26.2) | 0.74 |
| Atrial fibrillation–n (%) | 176 (44.2) | 331 (46.4) | 0.48 |
| Heart failure–n (%) | 169 (42.5) | 322 (45.2) | 0.40 |
| Chronic kidney disease–n (%)* | 63 (15.8) | 151 (21.2) | **0.03** |
| Chronic obstructive pulmonary disease–n (%) | 49 (12.3) | 101 (14.2) | 0.39 |
| History of cancer–n (%) | 96 (24.1) | 192 (26.9) | 0.31 |
| **Echocardiographic parameters before TAVR** | | | |
| LVEF, % | 53.1 ± 14.7 | 52.5 ± 15.5 | 0.50 |
| LV mass, g/m$^2$ | 128 ± 35 | 138 ± 40 | **<0.001** |
| LVEDD, mm | 49.0 ± 8.4 | 50.8 ± 8.4 | **0.001** |
| LVESD, mm | 33.4 ± 9.1 | 35.6 ± 10.0 | **<0.001** |
| Mean aortic pressure gradient, mmHg | 46.6 ± 14.6 | 47.1 ± 14.7 | 0.58 |
| AVAi, cm$^2$/m$^2$ | 0.76 ± 0.24 | 0.75 ± 0.24 | 0.44 |
| Systolic PAP, mmHg | 40.7 ± 14.8 | 42.1 ± 14.4 | 0.15 |
| **Prehospital antithrombotic management** | | | |
| Dual APT–n (%) | 99 (24.9) | 159 (22.3) | 0.32 |
| Anticoagulant therapy–n (%) | 152 (337.9) | 304 (42.6) | 0.14 |
| **Discharge antithrombotic medication** | | | |
| Aspirin–n (%) | 360 (90.5) | 665 (93.3) | 0.09 |
| Clopidogrel–n (%) | 230 (57.8) | 367 (51.5) | **0.031** |
| Dual APT–n (%) | 223 (56.0) | 364 (51.1) | 0.091 |
| Anticoagulant therapy–n (%) | 165 (41.5) | 342 (48.0) | **0.041** |

Otherwise specified, data are presented as mean ± SD.

* Chronic kidney disease was defined by a creatinine level >150 μmol/L.

*Abbreviations*: APT = antiplatelet therapy; AVAi = aortic valve area indexed; BMI = body mass index; H-MPV = high mean platelet volume; L-MPV = low mean platelet volume; LV = left ventricle; LVEDD = left ventricular end-diastolic diameter; LVEF = left ventricle ejection fraction; LVESD = left ventricular end-systolic diameter; MPV = mean platelet volume; PAP = pulmonary arterial pressure; STS = society of thoracic surgeons; TAVR = transcatheter aortic valve replacement.

HMW vWF defect on MPV could be evidenced. By multivariate analysis, the independent predictors for baseline L-MPV were hemoglobin level (odds ratio (OR) 0.91, 95% CI [0.83–1.00], p = 0.045), platelets count (OR 1.93, 95% CI [1.58–2.36], p<0.001) and BNP (OR 0.97, 95% CI [0.95–0.99], p = 0.001).

**Table 2. Biological findings prior to TAVR.**

| Variables | MPV before TAVR | | p value |
|---|---|---|---|
| | ≤10 fL L-MPV (n = 398) | >10 fL H-MPV (n = 713) | |
| Hemoglobin–g/dL | 11.9 ± 1.7 | 12.1 ± 1.7 | **0.016** |
| Platelets–G/L | 249 ± 78 | 215 ± 69 | **<0.001** |
| MPV/PC ratio | 4.2 ± 1.5 | 5.8 ± 2.9 | **<0.001** |
| Leukocytes–G/L | 7.6 ± 2.4 | 7.5 ± 2.4 | 0.55 |
| Creatinine–μmol/L | 113 ± 64 | 121 ± 80 | 0.055 |
| eGFR–mL/min | 55 ± 21 | 53 ± 21 | 0.065 |
| BNP–pg/mL | 532 ± 780 | 753 ± 996 | **<0.001** |
| CT-ADP | 191 ± 81 | 189 ± 77 | 0.74 |
| PT–% | 81 ± 20 | 80 ± 19 | 0.59 |

Data are expressed as mean ± SD.

*Abbreviations*: BNP = B-type natriuretic peptide; CT-ADP = closure time adenosine diphosphate; eGFR = estimated glomerular filtration rate; H-MPV = high mean platelet volume; L-MPV = low mean platelet volume; MPV = mean platelet volume; PC = platelet count; PT = prothrombin time; TAVR = transcatheter aortic valve replacement.

## Discussion

The current study drawn from a cohort of 1,111 patients who underwent TAVR specifically aimed to assess the prognostic impact of baseline MPV on postprocedural bleeding and thrombotic events. The salient results of the present study are as follows: (i) low MPV at baseline is an independent predictor of early and late bleeding events following TAVR; (ii) this

**Table 3. Outcomes during first year after TAVR.**

| Variables | MPV before TAVR | | p value |
|---|---|---|---|
| | ≤10 fL L-MPV (n = 398) | >10 fL H-MPV (n = 713) | |
| **Bleeding events** | | | |
| Major and life-threatening bleeding <1 year | 91 (22.9) | 126 (17.7) | **0.034** |
| Major and life-threatening bleeding <30 days | 82 (20.6) | 115 (16.1) | 0.057 |
| Major bleeding <30 days | 62 (15.6) | 67 (9.4) | **<0.01** |
| Major bleeding between 30 days and 1 year | 21 (5.3) | 30 (4.2) | 0.41 |
| Life threatening bleeding <30 days | 28 (7.0) | 54 (7.6) | 0.75 |
| Life-threatening between 30 days and 1 year | 11 (2.8) | 15 (2.1) | 0.48 |
| Bleeding requiring RBC transfusion >2 U | 95 (23.9) | 125 (17.5) | **0.01** |
| Minor bleeding | 74 (18.6) | 126 (17.7) | 0.68 |
| Access site vascular complications | 144 (36.2) | 206 (28.9) | **0.012** |
| **Mortality** | | | |
| Death from any cause | 60 (15.1) | 104 (14.6) | 0.84 |
| Cardiovascular death | 38 (9.5) | 58 (8.1) | 0.41 |
| **Ischemic events** | | | |
| Myocardial infarction | 14 (3.5) | 18 (2.5) | 0.34 |
| Stroke | 35 (8.8) | 59 (8.3) | 0.77 |

Data are expressed as n (%).

*Abbreviations*: H-MPV = high mean platelet volume; L-MPV = low mean platelet volume; MPV = mean platelet volume; RBC = red blood cells; TAVR = transcatheter aortic valve replacement.

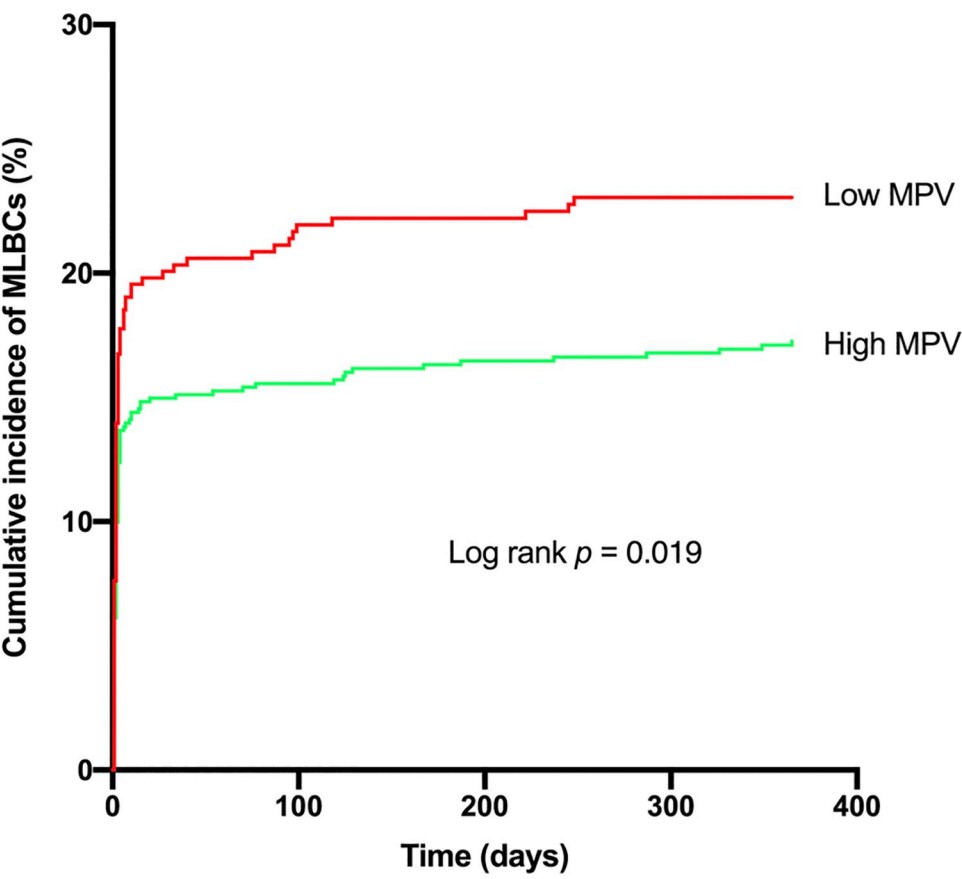

**Fig 2. Impact of baseline mean platelet volume on the incidence of major/life-threatening bleeding complications at one year follow-up after TAVR.** This figure shows the cumulative incidence of MLBCs during the first year after TAVR according to baseline MPV. Low MPV was defined as a baseline MPV ≤10 fL and high MPV as a baseline MPV >10 fL. *Abbreviations*: MLBCs = major and life-threatening bleeding complications; MPV = mean platelet volume; TAVR = transcatheter aortic valve replacement.

effect is independent of the extent of platelet P2Y12 inhibition and HMW-multimers defect of vWF; (iii) no impact of MPV on thrombotic events including stroke and myocardial infarction could be evidenced. Our findings suggest that preprocedural low MPV is an independent risk factor for bleeding after TAVR independently of ongoing antithrombotic therapy and HMW vWF-related primary hemostasis disorders.

## Pivotal role of platelets in primary haemostasis after TAVR

Numerous studies have investigated the impact of platelets in the tight regulation and balance of bleeding and thrombotic risk factors in TAVR. Previous works have emphasized that thrombocytopenia occurred in 40 to 70% of patients undergoing TAVR [4,5] and is associated with a higher prevalence of hypoattenuated leaflet thickening, a surrogate marker of valve thrombosis [6], enhanced stroke risk [7] and increased mortality [5]. More recently, data from the Japanese multicenter OCEAN-TAVI registry have underlined that thrombocytopenia is an important predictor of increased bleeding risk and mortality [8]. Accordingly, chronic thrombocytopenia was associated with increased periprocedural complications in the field of interventional cardiovascular procedures such as TAVR and MitraClip [18]. Although the full effect and mechanisms of thrombocytopenia have not yet been fully elucidated, the swift

**Table 4.** Univariate and multivariate analyses for occurrence of major/life-threatening bleeding during the first year after TAVR.

| Variables | Univariate analysis | | Multivariate analysis* | |
|---|---|---|---|---|
| | sHR (95% CI) | p value | sHR (95% CI) | p value |
| Age–years | 1.02 (1.00–1.04) | **0.03** | 1.02 (0.99–1.05) | 0.17 |
| Male sex | 0.68 (0.52–0.89) | **0.004** | 0.88 (0.62–1.23) | 0.45 |
| BMI–kg/m$^2$ | 0.98 (0.96–1.01) | 0.14 | | |
| BMI >30kg/m$^2$ | 1.01 (0.75–1.38) | 0.93 | | |
| Logistic EuroSCORE–% | 1.01 (1.00–1.02) | **0.002** | 1.01 (1.00–1.03) | **0.03** |
| EuroSCORE 2 –% | 1.01 (1.00–1.03) | 0.12 | | |
| STS mortality–% | 1.03 (1.02–1.04) | **<0.001** | 1.01 (0.98–1.03) | 0.60 |
| Current smoking | 0.87 (0.41–1.83) | 0.70 | | |
| Hypertension | 1.15 (0.80–1.66) | 0.45 | | |
| Dyslipidemia | 0.89 (0.68–1.16) | 0.37 | | |
| Diabetes mellitus | 0.96 (0.72–1.27) | 0.75 | | |
| Coronary artery disease | 1.14 (0.88–1.49) | 0.31 | | |
| Peripheral artery disease | 0.94 (0.70–1.27) | 0.69 | | |
| Atrial fibrillation | 1.31 (1.01–1.71) | **0.04** | 1.28 (0.80–2.04) | 0.31 |
| Heart failure | 1.17 (0.90–1.52) | 0.23 | | |
| Chronic kidney disease | 1.00 (0.70–1.38) | 0.99 | | |
| COPD | 0.63 (0.39–0.97) | **0.04** | 0.74 (0.42–1.27) | 0.27 |
| History of cancer | 1.17 (0.87–1.56) | 0.30 | | |
| Dual APT on admission | 0.80 (0.58–1.11) | 0.18 | | |
| Anticoagulant on admission | 1.18 (0.91–1.54) | 0.21 | | |
| Transfemoral approach | 0.98 (0.61–1.57) | 0.92 | | |
| Transcarotid approach | 0.54 (0.24–1.21) | 0.13 | | |
| Balloon aortic valvuloplasty | 1.32 (0.83–2.10) | 0.23 | | |
| Valve size | 0.97 (0.92–1.02) | 0.18 | | |
| Sheath size–F | 1.04 (0.97–1.11) | 0.31 | | |
| Sapien | 0.88 (0.68–1.15) | 0.36 | | |
| CoreValve | 1.15 (0.89–1.50) | 0.29 | | |
| Acurate | 2.00 (0.26–15.62) | 0.51 | | |
| Post-dilatation | 1.06 (0.71–1.58) | 0.78 | | |
| Aspirin at discharge[‡] | 0.48 (0.31–0.74) | **<0.001** | | |
| Clopidogrel at discharge | 0.77 (0.59–1.00) | 0.05 | | |
| Dual APT at discharge | 0.73 (0.56–0.96) | **0.02** | 0.65 (0.41–1.04) | 0.07 |
| Anticoagulant at discharge | 1.19 (0.91–1.55) | 0.21 | | |
| Hemoglobin at baseline–g/dL | 0.80 (0.74–0.86) | **<0.001** | 0.84 (0.75–0.93) | **0.001** |
| Platelets at baseline per 100 G/L | 1.00 (1.00–1.00) | **0.03** | 0.78 (0.59–1.02) | 0.07 |
| MPV ≤10 fL at baseline | 1.33 (1.03–1.74) | **0.03** | 1.64 (1.16–2.32) | **0.005** |
| MPV/PC ratio at baseline | 1.02 (0.99–1.07) | 0.15 | | |
| Leukocytes at baseline–G/L | 0.97 (0.91–1.03) | 0.33 | | |
| Creatinine at baseline–μmol/L | 1.00 (1.00–1.00) | 0.82 | | |
| eGFR at baseline–mL/min | 0.99 (0.99–1.00) | 0.08 | | |
| BNP at baseline[†] –pg/mL | 1.01 (1.00–1.03) | 0.05 | | |
| TP at baseline–% | 1.00 (0.99–1.00) | 0.50 | | |
| CT-ADP at baseline–sec | 1.00 (1.00–1.00) | 0.09 | | |
| CT-ADP post-TAVR >180sec | 2.68 (1.94–3.70) | **<0.001** | 2.72 (1.87–3.95) | **<0.001** |

(*Continued*)

**Table 4.** (Continued)

| Variables | Univariate analysis | | Multivariate analysis* | |
|---|---|---|---|---|
| | sHR (95% CI) | p value | sHR (95% CI) | p value |
| PRI-VASP post-TAVR | 0.99 (0.98–1.00) | 0.10 | | |

*Calculated using a Fine and Gray model including all univariate predictors at p <0.05.

† per 100 pg/mL.

‡ The variable Aspirin at discharge was not included in the multivariate analysis because of a high collinearity with DAPT at discharge.

*Abbreviations*: APT = antiplatelet therapy; BMI = body mass index; BNP = B-type natriuretic peptide; CI = confidence interval; COPD = chronic obstructive pulmonary disease; CT-ADP = closure time adenosine diphosphate; eGFR = estimated glomerular filtration rate; HR = hazard ratio; MPV = mean platelet volume; PC = platelet count; PRI = platelet reactivity index; PT = prothrombin time; sHR = sub-distribution hazards ratios; STS = society of thoracic surgeons; TAVR = transcatheter aortic valve replacement.

normalization of platelet count days after the procedure suggests that platelet consumption is likely the consequence of shear induced platelet activation and aggregation and/or interactions between platelets, polyethylene terephthalate and pericardial wraps. Other reports have underlined that thrombocytopenia occurred more frequently with balloon-expandable devices [19]. Animal models provided evidence that platelets produced after thrombocytopenia not only had a higher mean platelet volume but were also more reactive and enriched in thromboxane A2 [9].

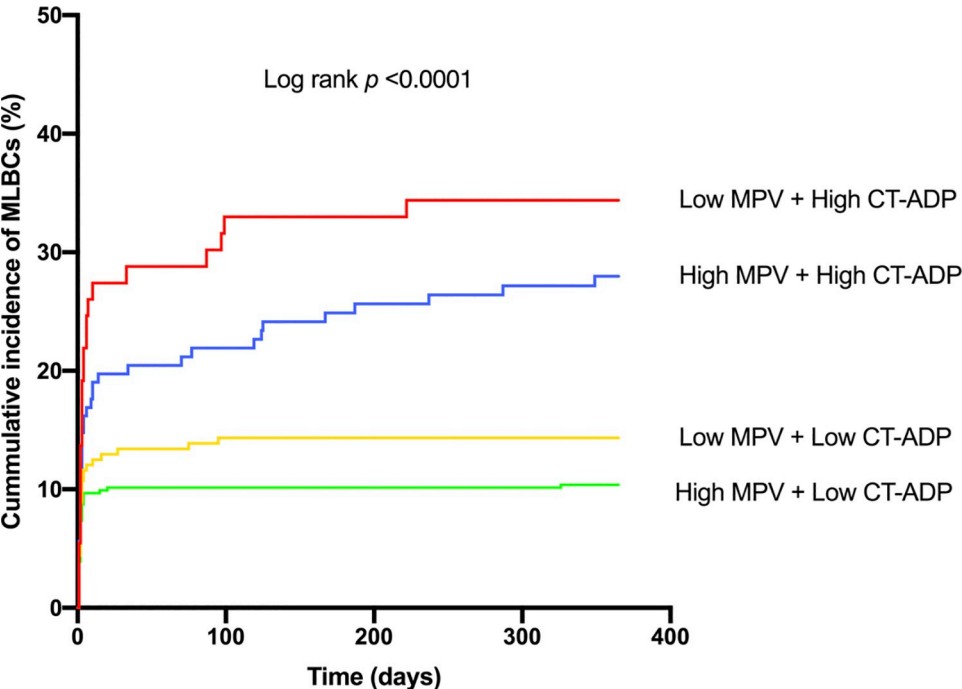

**Fig 3. Impact of baseline mean platelet volume and postprocedural CT-ADP on the incidence of major/life-threatening bleeding complications at one year follow-up.** This figure shows the impact of baseline MVP and postprocedural CT-ADP value on the incidence of major/life-threatening bleedings. Patients were stratified according to baseline MPV (threshold 10 fL) and postprocedural CT-ADP value (threshold 180 s). *Abbreviations*: CT-ADP = closure time adenosine diphosphate; MPV = mean platelet volume; TAVR = transcatheter aortic valve replacement.

**Table 5. Univariate and multivariate logistic regression analyses for mean platelet volume ≤10 fL at baseline.**

| Variables | Univariate analysis | | Multivariate analysis | |
|---|---|---|---|---|
| | OR (95% CI) | p value | OR (95% CI) | p value |
| Age–years | 0.99 (0.98–1.01) | 0.47 | | |
| Male sex | 0.99 (0.77–1.27) | 0.94 | | |
| BMI >30 kg/m$^2$ | 0.86 (0.64–1.16) | 0.33 | | |
| Logistic EuroSCORE–% | 0.99 (0.98–1.00) | 0.06 | | |
| EuroSCORE 2 –% | 0.99 (0.97–1.01) | 0.35 | | |
| STS mortality–% | 1.00 (0.98–1.02) | 0.93 | | |
| Current smoking | 0.93 (0.48–1.79) | 0.82 | | |
| Hypertension | 0.85 (0.62–1.18) | 0.33 | | |
| Dyslipidemia | 0.94 (0.73–1.20) | 0.60 | | |
| Diabetes mellitus | 1.11 (0.85–1.44) | 0.45 | | |
| Coronary artery disease | 1.18 (0.92–1.51) | 0.19 | | |
| Peripheral artery disease | 1.05 (0.79–1.38) | 0.74 | | |
| Atrial fibrillation | 0.91 (0.72–1.17) | 0.48 | | |
| Heart failure | 0.90 (0.70–1.15) | 0.40 | | |
| Chronic kidney disease | 0.70 (0.51–0.97) | **0.03** | 0.83 (0.56–1.24) | 0.36 |
| Chronic obstructive pulmonary disease | 0.85 (0.59–1.23) | 0.39 | | |
| History of cancer | 0.86 (0.65–1.14) | 0.31 | | |
| Dual APT on admission | 1.16 (0.87–1.54) | 0.33 | | |
| Anticoagulant on admission | 0.83 (0.65–1.07) | 0.14 | | |
| Hemoglobin–g/dL | 0.91 (0.85–0.98) | **0.02** | 0.91 (0.83–1.00) | **0.04** |
| Platelets per 100 G/L | 1.90 (1.59–2.26) | **<0.001** | 1.93 (1.58–2.36) | **<0.001** |
| Leukocytes–G/L | 1.02 (0.97–0.98) | 0.54 | | |
| Creatinine–μmol/L | 1.00 (1.00–1.00) | 0.09 | | |
| eGFR–mL/min | 1.01 (1.00–1.01) | 0.07 | | |
| BNP at baseline[†] –pg/mL | 0.97 (0.95–0.99) | **<0.001** | 0.97 (0.95–0.99) | **<0.001** |
| CT-ADP–sec | 1.00 (1.00–1.00) | 0.74 | | |
| PT–% | 1.0 (1.00–1.00) | 0.60 | | |
| LVEF–% | 1.31 (0.58–2.95) | 0.51 | | |
| Mean aortic gradient–mmHg | 1.00 (0.99–1.01) | 0.55 | | |
| Aortic valve area–cm$^2$ | 1.23 (0.73–2.07) | 0.43 | | |

[†] per 100 pg/mL.

*Abbreviations*: APT = antiplatelet therapy; BMI = body mass index; BNP = B-type natriuretic peptide; CI = confidence interval; CT-ADP = closure time adenosine diphosphate; eGFR = estimated glomerular filtration rate; LVEF = left ventricle ejection fraction; MPV = mean platelet volume; OR = odds ratio; PC = platelet count; PT = prothrombin time.

## Low mean platelet volume as a surrogate marker of bleeding events after TAVR

With respect to other biomarkers of platelet reactivity that require expensive and time consuming laboratory analyses, MPV is a simple, common, automatically generated and valuable measurement routinely assessed as part of the basic workflow in hemostasis labs [10,11]. Owing the key role of platelets in the time course of the thrombotic process, numerous studies have explored the impact of MPV on cardiovascular and metabolic disorders. High MPV has been observed in patients with cardiovascular risk factors, such as diabetes, dyslipidemia, obesity, smoking and hypertension [20–24]. In line with these observations, high MPV was demonstrated to be a potent predictor of recurrent thrombotic events including myocardial

infarction and stroke [25–27]. By contrast, lower values of MPV have been associated with recurrent epistaxis [28] and Henoch-Schonlein purpura activity [29].

During TAVR, pioneering work by Magri et al. has suggested that low MPV may be indicative of a higher bleeding risk [14]. However, in the absence of systematic capture of key determinants of bleeds such as P2Y12 inhibition or the loss of HMW-multimers defect of vWF, the specific contribution of MPV on adverse events was difficult to ascertain. In our study, we identified MPV as a predictor of MLBCs after TAVR independently of the extent of P212 inhibition by thienopyridines or the magnitude of vWF defect. Patients with lower values, defined by baseline MPV ≤10 fL, had a higher bleeding risk during the first year after the procedure. Our results challenge a previous study of very limited size that reported enhanced hemorrhagic complications among high MPV patients after TAVR [13].

While the precise description of the pathophysiological mechanisms linking high MPV and platelets activity are far beyond the scope of the present study, several hypotheses could be raised. Larger platelets were described to contain more prothrombotic molecules such as thromboxane A2, P-selectin, and platelet-derived growth factor [10] together with higher glycoprotein IIb-IIIa receptor expression [30]. Consistently, larger platelets were found more frequently reticulated and less sensitive to antiplatelet therapy [31]. In line with this paradigm, we have recently demonstrated that the incidence of early major bleeding events was significantly enhanced among patients with adequate P2Y12 inhibition by clopidogrel [17]. During TAVR, early bleeds are mainly the consequence of periprocedural vascular complications [2]. Vascular injury exposes the subendothelial matrix enriched in thrombotic factors to the blood flow leading to subsequent platelet adhesion and activation through vWF and GPIb dependent mechanisms. While excessive activation of this process may lead to thrombosis, it is likely that platelets with high MPV could be more active and therefore more prone to prompt vascular wound healing. Importantly, this protective action of high MPV was not counterbalanced by an excess rate of thrombotic events after TAVR such as stroke and myocardial infarction.

## Determinants of mean platelet volume

Several cardiovascular risk factors (e.g. smoking or obesity), hemostasis parameters (e.g. platelet count) and clinical status (e.g. heart failure) were described to influence MPV [32]. Indeed, high MPV was associated with CKD, high haemoglobin level, low platelet count and high BNP level in our work. Sympathetic activation, hemodynamic factors, vascular changes, oxidative stress and inflammation were proposed to explain the extent of platelet activation during heart failure [33]. Consistently, significant drop of MPV and BNP level have been described after TAVR as a possible result of hemodynamic improvement and blunt platelet activation [34].

## Clinical implications

Previous studies have stressed that MLBCs occurred in up to 20% of TAVR procedures performed in severe AS patients [2,35], consistent with the 19.5% rate observed in our cohort. Both early and late bleeding complications independently concur to increase all-cause mortality [2,12,36]. Among various factors that were evidenced to contribute to enhanced bleeding risk, initial attention was given to the role played by paravalvular leak, ongoing flow turbulence and persistent HMW vWF defect. We and others have demonstrated that CT-ADP >180 s, as a surrogate marker of ongoing HMW is a strong predictor of MLBCs. The present data underline that low MPV, independently to CT-ADP values contribute to enhance bleeds. Giving the adverse impact of bleeding on prognosis, the optimal antithrombotic regimen after successful TAVR remains a subject of intense interest and the benefit of de-escalation strategies highlighted by several trials including the POPULAR study [37]. Owing to the rapid extension

of its indications, TAVR will concern patients with heterogeneous profiles, bleeding and thrombotic risks and tailored antithrombotic strategies will be of paramount importance [38]. In that setting, there is a need to identify markers able to assess with precision the difficult homeostasis balance between thrombotic and bleeding risk occurring after the procedure [1]. Among various candidates, baseline MPV should be regarded as a simple, common and inexpensive biomarker of particular interest for the multiparametric assessment of bleeding risk following TAVR.

## Study limitations

This study has several limitations. First, the observational nature of this work should be acknowledged and do not allow to draw any definite conclusions about the pathophysiological role of activated platelets in bleeding events and vascular complications after TAVR. Other platelet reactivity markers, such as P-selectin, glycoprotein (Gp) V, platelet-derived microparticles, growth differentiation factor 15 (GDF-15) were not analyzed in this study. MPV should be viewed as a marker of bleeding risk following TAVR pending further investigations. Second, the MPV threshold used to differentiate L-MPV and H-MPV depends on the laboratory measurement method and therefore cannot be generalized. However, the threshold of 10 fL used in the present study is very similar to the one originally described in the work from Magri et al [14] and those depicted in previous reports [10]. Third, the time course of MPV during the procedure was not examined.

## Conclusion

MPV is a simple, common and automatically generated measurement of platelet size. Baseline low MPV is a strong predictor of early and late MLBCs following TAVR and irrespective of the extent of platelet P2Y12 inhibition and HMW vWF-related primary hemostasis disorders. MPV should be regarded as an easy and valuable biomarker of bleeding prediction after TAVR.

## Supporting information

**S1 Table. Procedural characteristics.**
(DOCX)

**S2 Table. Patients with and without MPV ≤10 fL in the matched cohort for dual antiplatelet and anticoagulant therapies.**
(DOCX)

**S3 Table. Sensitivity analyses assessing association of major and life-threatening bleeding complications during the first year after TAVR with MPV ≤10 fL.**
(DOCX)

**S4 Table. Outcomes during first year after TAVR in the subgroup of patients treated with new generation valves (Sapien 3, Evolut-Pro and Accurate-Neo).**
(DOCX)

## Author Contributions

**Conceptualization:** Antonin Trimaille, Olivier Morel.

**Data curation:** Antonin Trimaille, Kensuke Matsushita, Benjamin Marchandot, Adrien Carmona, Sébastien Hess, Marion Kibler, Joé Heger, Antje Reydel, Laurent Sattler, Lelia Grunebaum, Laurence Jesel, Patrick Ohlmann, Olivier Morel.

**Formal analysis:** Antonin Trimaille, Kensuke Matsushita, Benjamin Marchandot, Adrien Carmona, Sébastien Hess, Marion Kibler, Joé Heger, Antje Reydel, Laurent Sattler, Lelia Grunebaum, Laurence Jesel, Patrick Ohlmann, Olivier Morel.

**Investigation:** Antonin Trimaille, Kensuke Matsushita, Benjamin Marchandot, Adrien Carmona, Sébastien Hess, Marion Kibler, Joé Heger, Antje Reydel, Laurent Sattler, Lelia Grunebaum, Laurence Jesel, Patrick Ohlmann, Olivier Morel.

**Methodology:** Antonin Trimaille, Kensuke Matsushita, Olivier Morel.

**Resources:** Olivier Morel.

**Software:** Kensuke Matsushita, Olivier Morel.

**Supervision:** Olivier Morel.

**Validation:** Laurent Sattler, Lelia Grunebaum, Laurence Jesel, Patrick Ohlmann, Olivier Morel.

**Visualization:** Olivier Morel.

**Writing – original draft:** Antonin Trimaille, Olivier Morel.

**Writing – review & editing:** Antonin Trimaille, Kensuke Matsushita, Benjamin Marchandot, Olivier Morel.

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
