## [Decision Letter · Decision Letter 0]

1 Oct 2021

PONE-D-21-27214Baseline Mean Platelet Volume Is a Strong Predictor of Major and Life-threatening Bleedings After Transcatheter Aortic Valve ReplacementPLOS ONE

Dear Authors,

Thank you for submitting your manuscript to PLOS ONE. After careful consideration, we feel that it has merit but does not fully meet PLOS ONE’s publication criteria as it currently stands. Therefore, we invite you to submit a revised version of the manuscript that addresses the points raised during the review process.

Please revise your manuscript according to the comments by the Reviewers.

We look forward to receiving your revised manuscript.

Kind regards,

Alberto Aimo, MD

Academic Editor

PLOS ONE

Journal Requirements:

https://journals.plos.org/plosone/s/file?id=ba62/PLOSOne_formatting_sample_title_authors_affiliations.pd

Reviewers' comments:

Reviewer's Responses to Questions

**Comments to the Author**

1. Is the manuscript technically sound, and do the data support the conclusions?

Reviewer #1: Yes

Reviewer #2: Partly

2. Has the statistical analysis been performed appropriately and rigorously? 

Reviewer #1: Yes

Reviewer #2: No

3. Have the authors made all data underlying the findings in their manuscript fully available?

Reviewer #1: Yes

Reviewer #2: No

4. Is the manuscript presented in an intelligible fashion and written in standard English?

Reviewer #1: Yes

Reviewer #2: Yes

5. Review Comments to the Author

Reviewer #1: Author reported a retrospective analysis of patients treated with TAVI, where they analysed the impact of different baseline MPV on bleeding and vascular outcomes.

They found that MPV is an independent predictor if MLBCs after TAVI.

The topic is interesting, considering that vascular complications actually are the major complications of TAVI.

However, authors in method section reported that they included patients treated between February 2010

and May 2019. Then they reported that valve used are Edward Sapien XT and S3 and Evolut R/PRO, and also Accurate Neo.

It has been described as new generation device (S3 vs XT, Evolut R/PRO vs Corevalve) had reduced vascular complications due to small profile introducer. So, considering that in Europe Evolut R was introduced in 2014, there is the possibility that some part of patients treated with self-expandable valve before 2014 was treated with Corevalve, that required a 18F sheat, as Sapien XT.

My suggestion is to perform a subanalyses of this patients including only patients treated with low profile valve (Evolut R/PRO, S3, and Accurate Neo), in order to asses if the results are confirmed with the low profile devices.

Moreover, it could be preferable to report the size of the sheat used when you talk about vascular complications and bleeding.

Reviewer #2: Title of the study: “Baseline Mean Platelet Volume Is a Strong Predictor of Major and Life-threatening Bleedings After Transcatheter Aortic Valve Replacement”.

In the present study the authors aimed to assess the prognostic significance of baseline mean platelet volume on bleeding events after TAVR. At 1-year follow-up, patients with low MPV (36%) had met more often the endpoint of major/life-threatening bleeding complications, with no differences as for death or ischemic events. Low MPV was an independent predictor of the primary endpoint at multivariable analysis.

Comments to the authors:

- Please verify that each acronym is defined when firstly reported in the text (e.g., CT-ADP in the abstract).

- The introduction section seems confusing, a careful revision and shortening is advised, clarifying the objectives and the importance of the analysis performed.

- It should be clarified whether aspirin, and not clopidogrel, was administered to all patients on oral anticoagulants and the reasons behind this choice.

- Pre-TAVR and discharge therapies of the two groups should be reported in table 1. In case of any difference, it could be advisable to perform a sensitivity analysis by harmonizing the two groups for antiplatelet/anticoagulants therapies.

- Collinearity among the variables included in the regression analysis should be assessed.

- Please verify the HR and 95%CI reported for platelets at baseline.

- Have not normally distributed variables been ln-transformed before entering regression models?

- Units of measure should be reported for each variable in Table 4 and 5.

- How were variables selected to entering the regression models?

6. PLOS authors have the option to publish the peer review history of their article (what does this mean?). If published, this will include your full peer review and any attached files.

Reviewer #1: No

Reviewer #2: No

---

## [Author Response · Author response to Decision Letter 0]

24 Oct 2021

Rebuttal letter – Manuscript PONE-D-21-27214

“Baseline Mean Platelet Volume Is a Strong Predictor of Major and Life-threatening Bleedings After Transcatheter Aortic Valve Replacement”

We thank the Editorial Committee of Plos One and the Referees for their relevant feedback that allowed us to substantially improve the manuscript. 

Please find below the point-by-point response to the Editorial committee and Reviewers comments.

 

Journal Requirements

3. Please include captions for your Supporting Information files at the end of your manuscript, and update any in-text citations to match accordingly. 

We have taken these points into consideration and amended the manuscript accordingly.

 

Reviewers' comments

Reviewer #1

Author reported a retrospective analysis of patients treated with TAVI, where they analysed the impact of different baseline MPV on bleeding and vascular outcomes. They found that MPV is an independent predictor of MLBCs after TAVI. The topic is interesting, considering that vascular complications actually are the major complications of TAVI.

However, authors in method section reported that they included patients treated between February 2010 and May 2019. Then they reported that valve used are Edward Sapien XT and S3 and Evolut R/PRO, and also Accurate Neo. It has been described as new generation device (S3 vs XT, Evolut R/PRO vs Corevalve) had reduced vascular complications due to small profile introducer. So, considering that in Europe Evolut R was introduced in 2014, there is the possibility that some part of patients treated with self-expandable valve before 2014 was treated with Corevalve, that required a 18F sheat, as Sapien XT.

My suggestion is to perform a subanalyses of this patients including only patients treated with low profile valve (Evolut R/PRO, S3, and Accurate Neo), in order to asses if the results are confirmed with the low profile devices. Moreover, it could be preferable to report the size of the sheat used when you talk about vascular complications and bleeding.

We would like to thank the Reviewer 1 for her/his appreciation of our work and for her/his very relevant suggestion. 

According to Reviewer’s request, we performed a sub-analyze on patients treated with low risk profile valves (Sapien 3, Evolut-Pro and Accurate-Neo). We observed a significant reduction of major bleeding <30 days in patients with H-MPV in comparison with L-MPV. Other bleeding events were also reduced in H-MPV patients although the significance was not reached because owing to the limited size of this sub-group.

We added the analyses in the Supplemental Table 4 and we amended the main text as follows:

Page 11, Lines 5-7: “Since reduced vascular complications were described with new generation devices (Sapien 3, Evolut-Pro and Acurate-Neo), additional subanalyses were performed on patients treated with these devices.”

Page 13, Lines 7-13: “Finally, we performed a subanalysis on the 618 patients treated with new generation valves (Sapien 3, Evolut-Pro and Accurate-Neo). Among them, 217 (35.1%) patients had L-MPV at baseline and 401 (64.9%) had H-MPV. While major bleeding <30 days was significantly lower in H-MPV patients in comparison with L-MPV patients (9.0% vs 15.7%, p=0.022), MLBCs at one-year follow-up did not reach significance difference (17.2% vs 22.1%, p=0.27) probably due to the limited size of this sub-group (Table S4).”

In addition, we added the sheat size in the Supplemental Table 1. We did not observe any difference between patients with low MPV and patients with high MPV.

Table S1. Procedural characteristics.

Variables MPV before TAVR p value

 ≤10 fL

L-MPV

(n = 398) >10 fL

H-MPV

(n = 713) 

Transfemoral approach 367 (92.2) 650 (91.2) 0.55

Transcarotid approach 17 (4.3) 36 (5.0) 0.56

Sheat size 

14 F 219 (55.0) 407 (57.1) 0.44

16 F 68 (17.1) 140 (19.6) 0.27

18 F 91 (22.9) 131 (18.4) 0.14

20 F or more 18 (4.5) 27 (3.8) 0.82

Balloon aortic valvuloplasty 34 (8.5) 43 (6.0) 0.11

Sapien 240 (60.3) 429 (60.2) 0.95

CoreValve 155 (38.9) 285 (40.0) 0.75

Acurate 2 (0.5) 1 (0.1) 0.29

Post-dilatation 47 (11.8) 87 (12.2) 0.47

Data are expressed as n (%).

Abbreviations: H-MPV = high mean platelet volume; L-MPV = low mean platelet volume; MPV = mean platelet volume; TAVR = transcatheter aortic valve replacement

 

Reviewer #2

Title of the study: “Baseline Mean Platelet Volume Is a Strong Predictor of Major and Life-threatening Bleedings After Transcatheter Aortic Valve Replacement”.

In the present study the authors aimed to assess the prognostic significance of baseline mean platelet volume on bleeding events after TAVR. At 1-year follow-up, patients with low MPV (36%) had met more often the endpoint of major/life-threatening bleeding complications, with no differences as for death or ischemic events. Low MPV was an independent predictor of the primary endpoint at multivariable analysis.

We would like to thank the Reviewer 2 for her/his in-depth review of our work. Please find below our responses to these important queries.

Comments to the authors:

- Please verify that each acronym is defined when firstly reported in the text (e.g., CT-ADP in the abstract).

As requested, we defined CT-ADP in the Abstract and checked abbreviations through the manuscript.

- The introduction section seems confusing, a careful revision and shortening is advised, clarifying the objectives and the importance of the analysis performed.

We have performed a comprehensive revision of the “Introduction” part of the main text to make it more understandable and shorter, and to clarify the objectives of our work:

Page 6, Lines 1-25 and Page 7, Lines1-13: “Bleeding events after transcatheter aortic valve replacement (TAVR) still occur in a sizeable proportion of patients and mitigate long-term benefits [1,2] despite a substantial reduction of vascular complications over the last decade. Several reports have demonstrated that major bleedings following TAVR outweighed ischemic complications and dramatically impacted long-term prognosis [3]. As such, bleeds have been identified as an important area for improved patient care. The optimal antithrombotic regimen after TAVR is not clearly defined. Thrombotic and bleeding risk assessments in TAVR patients are complex and individually tailored antithrombotic therapy according to a specific risk profile are mandatory.

In the search of accurate biomarkers of bleeding risk after TAVR, several studies have investigated the impact of platelets. Thrombocytopenia occurred in 40 to 70% of patients undergoing TAVR [4,5] and is significantly associated with increased mortality [5], thrombosis events such as subclinical leaflet thrombosis [6] and stroke [7], and bleeding [8]. Moreover, it has been previously showed that platelets produced after thrombocytopenia were more reactive [9]. 

Mean platelet volume (MPV) has been proposed as a surrogate marker of platelet reactivity [10]. It is a machine-calculated measurement of the average platelet size that ranges in normal condition between 7 and 11 fL [11]. Owing to their enrichment in granules, vasoactive and prothrombotic factors, large platelets have been advocated as promising prothrombotic markers in various cardiovascular diseases and metabolic disorders [10]. Consistent with this paradigm, high MPV was associated with shortened bleeding time [9]. As previously demonstrated, increased shear stress during aortic stenosis (AS) is associated with a loss of the high molecular weight (HMW) of the von Willebrand factor (vWF) due to proteolysis of vWF multimers by the metalloproteinase ADAMTS13 [12]. Because large vWF multimers are stored in alpha-granules, low MPV may be the consequence of their release following flow disturbances. Conflicting evidence remain regarding the impact of MPV on bleeding and thrombotic risk in TAVR. Whilst an independent and paradoxical association between baseline high MPV and early bleeding after TAVR was evidenced in a limited cohort of patients [13], a larger trial later depicted a noxious impact of low MPV on periprocedural major and life-threatening bleeding complications (MLBCs) [14]. In the absence of systematic characterization of P2Y12 inhibition by clopidogrel or the assessment of vWF proteolysis, the independent contribution of MPV to bleeding events remains difficult to ascertain.

In the present study, we aimed to assess the independent predictive value of baseline MPV on bleeding complications after TAVR, and to investigate determinants of baseline MPV.

- It should be clarified whether aspirin, and not clopidogrel, was administered to all patients on oral anticoagulants and the reasons behind this choice.

This choice was based on the local protocol at the time of the study. To clarify, we have added this sentence in the “Methods” section of the main text:

Page 8, Lines 3-4: “According to the local protocol, only Aspirin was administered in patients with oral anticoagulant.”

- Pre-TAVR and discharge therapies of the two groups should be reported in table 1. In case of any difference, it could be advisable to perform a sensitivity analysis by harmonizing the two groups for antiplatelet/anticoagulants therapies.

According to Reviewer suggestion, we transferred data on preprocedural and discharges therapies from Supplemental Table 1 to Table 1.

We also performed the sensitivity analyses by adjusting for DAPT and anticoagulant therapy at discharge, which are known to impact major life-threatening bleeding events. The magnitude of MPV ≤10 fL was confirmed with additional sensitivity analyses adjusting for DAPT and anticoagulant therapy (sHR: 1.34; 95% CI: 0.98 to 1.84; p=0.07) (Table S3) but was not statistically significant because of the smaller sample size.

We amended the manuscript as follows: 

Page 10, Lines 23-25 and Page 11, Lines 1-5: “Additional sensitivity analyses for MPV ≤10 fL were performed with DAPT and anticoagulant therapy, which are known to impact MLBCs. Propensity score matching was performed to adjust DAPT and anticoagulant therapy between patients with and without MPV ≤10 fL. The propensity score model was developed using logistic regression and the model included DAPT and anticoagulant therapy as covariates. A nearest-neighbor algorithm was used to match patients with and without MPV ≤10 fL in a 1:1 ratio, with a caliper width equal to 0.2 of the SD of the logit of the propensity score.”

Page 13, Lines 1-6: “Results of our sensitivity analyses are shown in Tables S2 and S3. After propensity score matching, 389 patients with L-MPV were compared with 389 patients with H-MPV. The magnitude of preprocedural L-MPV was confirmed with additional sensitivity analyses adjusting for DAPT and anticoagulant therapy (sHR: 1.34; 95% CI [0.98-1.84]; p=0.07) (Table S3) but was not statistically significant because of the smaller sample size.”

Table S2. Patients with and without MPV ≤10 fL in the matched cohort for dual antiplatelet and anticoagulant therapies.

Variables Baseline MPV p value

 ≤10 fL

L-MPV

(n = 389) >10 fL

H-MPV

(n = 389) 

Demographic characteristics

Age – years 82.7 ± 7.2 82.5 ± 7.4 0.67

Male sex – n (%) 175 (45) 185 (48) 0.52

BMI – kg/m2 26.5 ± 5.3 27.1 ± 6.1 0.21

Logistic EuroSCORE – % 18.2 ± 12.6 19.6 ± 14.3 0.18

EuroSCORE 2 – % 5.3 ± 6.0 6.0 ± 6.5 0.12

STS mortality – % 6.3 ± 6.4 6.6 ± 5.4 0.55

Cardiovascular risk factors

Current smoking – n (%) 14 (3.6) 20 (5.1) 0.38

Hypertension – n (%) 317 (81.5) 322 (82.8) 0.71

Dyslipidemia – n (%) 226 (58.1) 231 (59.4) 0.77

Diabetes mellitus – n (%) 127 (32.6) 121 (31.1) 0.70

Medical history

Coronary artery disease – n (%) 198 (50.9) 181 (46.5) 0.25

Peripheral artery disease – n (%) 105 (27.0) 104 (26.7) 1.00

Atrial fibrillation – n (%) 172 (44.2) 158 (40.6) 0.35

Heart failure – n (%) 164 (42.3) 166 (42.7) 0.97

Chronic kidney disease – n (%)* 61 (15.7) 78 (20.1) 0.13

Chronic obstructive pulmonary disease – n (%) 48 (12.3) 63 (16.2) 0.15

History of cancer – n (%) 95 (24.4) 98 (25.2) 0.87

Echocardiographic parameters before TAVR

LVEF, % 53.2 ± 14.6 52.0 ± 15.3 0.25

LV mass, g/m2 128 ± 35 140 ± 42 <0.001

LVEDD, mm 49.1 ± 8.5 50.5 ± 8.5 0.02

LVESD, mm 33.4 ± 9.1 35.3 ± 10.0 0.01

Mean aortic pressure gradient, mmHg 46.6 ± 14.5 47.4 ± 14.1 0.43

AVA, cm2 0.76 ± 0.24 0.74 ± 0.22 0.27

Systolic PAP, mmHg 40.6 ± 14.9 40.8 ± 13.6 0.87

Prehospital antithrombotic management

Dual APT – n (%) 99 (25.5) 95 (24.4) 0.78

Anticoagulant therapy – n (%) 149 (38.3) 138 (35.5) 0.46

Discharge antithrombotic medication 

Aspirin – n (%) 359 (92.3) 364 (93.6) 0.58

Clopidogrel – n (%) 230 (59.1) 231 (59.4) 1.00

Dual APT – n (%) 223 (57.3) 223 (56.3) 1.00

Anticoagulant therapy – n (%) 165 (42.4) 165 (42.4) 1.00

Otherwise specified, data are presented as mean ± SD.

* Chronic kidney disease was defined by a creatinine level >150 µmol/L

Abbreviations: APT = antiplatelet therapy; AVAi = aortic valve area indexed; BMI = body mass index; H-MPV = high mean platelet volume; L-MPV = low mean platelet volume; LV = left ventricle; LVEDD = left ventricular end-diastolic diameter; LVEF = left ventricle ejection fraction; LVESD = left ventricular end-systolic diameter; MPV = mean platelet volume; PAP = pulmonary arterial pressure; STS = society of thoracic surgeons; TAVR = transcatheter aortic valve replacement

 

Table S3. Sensitivity analyses assessing association of major and life-threatening bleeding complications during the first year after TAVR with MPV ≤10 fL.

 MLBCs

sHR (95% CI) p value

MPV ≤10 fL 

Unadjusted 1.33 (1.03 - 1.74) 0.03

Adjusted* 1.34 (0.98 - 1.84) 0.07

* Adjusted for DAPT and anticoagulants therapy at discharge.

Abbreviations: MLBCs = major and life-threatening bleeding complications, MPV = mean platelet volume.

- Collinearity among the variables included in the regression analysis should be assessed.

The collinearity among continuous variables is shown below. Since no strong collinearity was observed, we decided to include all the continuous variables in the multivariate analysis.

 Age Euroscore log STS mortality Baseline Hemoglobin Baseline platelets

Age 1 0.2053045166 0.1536328629 0.0118122702 0.0199836297

Euroscore log 0.2053045166 1 0.4536356724 -0.102070058 -0.004542394

STS mortality 0.1536328629 0.4536356724 1 -0.149548306 0.0653244287

Baseline Hemoglobin 0.0118122702 -0.102070058 -0.149548306 1 -0.06849452

Baseline platelets 0.0199836297 -0.004542394 0.0653244287 -0.06849452 1

Given the high collinearity with the use of aspirin or clopidogrel and DAPT at discharge (all the patients under DAPT received aspirin and 97% of them received clopidogrel), we included only DAPT at discharge, which could be a strong factor of bleeding events at the chronic phase, in the multivariate analysis. 

Table 4. Univariate and multivariate analyses for occurrence of major/life-threatening bleeding during the first year after TAVR. 

Variables Univariate analysis Multivariate analysis*

 sHR (95% CI) p value sHR (95% CI) p value

Age – years 1.02 (1.00 - 1.04) 0.03 1.02 (0.99 - 1.05) 0.17

Male sex 0.68 (0.52 - 0.89) 0.004 0.88 (0.62 - 1.23) 0.45

BMI – kg/m2 0.98 (0.96 - 1.01) 0.14 

BMI >30kg/m2 1.01 (0.75 - 1.38) 0.93 

Logistic EuroSCORE – % 1.01 (1.00 - 1.02) 0.002 1.01 (1.00 - 1.03) 0.03

EuroSCORE 2 – % 1.01 (1.00 - 1.03) 0.12 

STS mortality – % 1.03 (1.02 - 1.04) <0.001 1.01 (0.98 - 1.03) 0.60

Current smoking 0.87 (0.41 - 1.83) 0.70 

Hypertension 1.15 (0.80 - 1.66) 0.45 

Dyslipidemia 0.89 (0.68 - 1.16) 0.37 

Diabetes mellitus 0.96 (0.72 - 1.27) 0.75 

Coronary artery disease 1.14 (0.88 - 1.49) 0.31 

Peripheral artery disease 0.94 (0.70 - 1.27) 0.69 

Atrial fibrillation 1.31 (1.01 - 1.71) 0.04 1.28 (0.80 - 2.04) 0.31

Heart failure 1.17 (0.90 - 1.52) 0.23 

Chronic kidney disease 1.00 (0.70 - 1.38) 0.99 

COPD 0.63 (0.39 - 0.97) 0.04 0.74 (0.42 - 1.27) 0.27

History of cancer 1.17 (0.87 - 1.56) 0.30 

Dual APT on admission 0.80 (0.58 - 1.11) 0.18 

Anticoagulant on admission 1.18 (0.91 - 1.54) 0.21 

Transfemoral approach 0.98 (0.61 - 1.57) 0.92 

Transcarotid approach 0.54 (0.24 - 1.21) 0.13 

Balloon aortic valvuloplasty 1.32 (0.83 - 2.10) 0.23 

Valve size 0.97 (0.92 - 1.02) 0.18 

Sheath size – F 1.04 (0.97 - 1.11) 0.31 

Sapien 0.88 (0.68 - 1.15) 0.36 

CoreValve 1.15 (0.89 - 1.50) 0.29 

Acurate 2.00 (0.26 - 15.62) 0.51 

Post-dilatation 1.06 (0.71 - 1.58) 0.78 

Aspirin at discharge‡ 0.48 (0.31 - 0.74) <0.001 

Clopidogrel at discharge 0.77 (0.59 - 1.00) 0.05 

Dual APT at discharge 0.73 (0.56 - 0.96) 0.02 0.65 (0.41 - 1.04) 0.07

Anticoagulant at discharge 1.19 (0.91 - 1.55) 0.21 

Hemoglobin at baseline – g/dL 0.80 (0.74 - 0.86) <0.001 0.84 (0.75 - 0.93) 0.001

Platelets at baseline per 100 G/L 1.00 (1.00 - 1.00) 0.03 0.78 (0.59 - 1.02) 0.07

MPV ≤10 fL at baseline 1.33 (1.03 - 1.74) 0.03 1.64 (1.16 - 2.32) 0.005

MPV / PC ratio at baseline 1.02 (0.99 - 1.07) 0.15 

Leukocytes at baseline – G/L 0.97 (0.91 - 1.03) 0.33 

Creatinine at baseline – μmol/L 1.00 (1.00 - 1.00) 0.82 

eGFR at baseline – mL/min 0.99 (0.99 - 1.00) 0.08 

BNP at baseline† – pg/mL 1.01 (1.00 - 1.03) 0.05 

TP at baseline – % 1.00 (0.99 - 1.00) 0.50 

CT-ADP at baseline – sec 1.00 (1.00 - 1.00) 0.09 

CT-ADP post-TAVR >180sec 2.68 (1.94 - 3.70) <0.001 2.72 (1.87 - 3.95) <0.001

PRI-VASP post-TAVR 0.99 (0.98 - 1.00) 0.10 

*Calculated using a Fine and Gray model including all univariate predictors at p <0.05

† per 100 pg/mL

‡ The variable Aspirin at discharge was not included in the multivariate analysis because of a high collinearity with DAPT at discharge. 

Abbreviations: APT = antiplatelet therapy; BMI = body mass index; BNP = B-type natriuretic peptide; CI = confidence interval; COPD = chronic obstructive pulmonary disease; CT-ADP = closure time adenosine diphosphate; eGFR = estimated glomerular filtration rate; HR = hazard ratio; MPV = mean platelet volume; PC = platelet count; PRI = platelet reactivity index; PT = prothrombin time; sHR = sub-distribution hazards ratios; STS = society of thoracic surgeons; TAVR = transcatheter aortic valve replacement. 

Table 5. Univariate and multivariate logistic regression analyses for Mean Platelet Volume ≤10 fL at baseline.

Variables Univariate analysis Multivariate analysis

 OR (95% CI) p value OR (95% CI) p value

Age – years 0.99 (0.98 - 1.01) 0.47 

Male sex 0.99 (0.77 - 1.27) 0.94 

BMI >30 kg/m2 0.86 (0.64 - 1.16) 0.33 

Logistic EuroSCORE – % 0.99 (0.98 - 1.00) 0.06 

EuroSCORE 2 – % 0.99 (0.97 - 1.01) 0.35 

STS mortality – % 1.00 (0.98 - 1.02) 0.93 

Current smoking 0.93 (0.48 - 1.79) 0.82 

Hypertension 0.85 (0.62 - 1.18) 0.33 

Dyslipidemia 0.94 (0.73 - 1.20) 0.60 

Diabetes mellitus 1.11 (0.85 - 1.44) 0.45 

Coronary artery disease 1.18 (0.92 - 1.51) 0.19 

Peripheral artery disease 1.05 (0.79 - 1.38) 0.74 

Atrial fibrillation 0.91 (0.72 - 1.17) 0.48 

Heart failure 0.90 (0.70 - 1.15) 0.40 

Chronic kidney disease 0.70 (0.51 - 0.97) 0.03 0.83 (0.56 - 1.24) 0.36

Chronic obstructive pulmonary disease 0.85 (0.59 - 1.23) 0.39 

History of cancer 0.86 (0.65 - 1.14) 0.31 

Dual APT on admission 1.16 (0.87 - 1.54) 0.33 

Anticoagulant on admission 0.83 (0.65 - 1.07) 0.14 

Hemoglobin – g/dL 0.91 (0.85 - 0.98) 0.02 0.91 (0.83 - 1.00) 0.04

Platelets per 100 G/L 1.90 (1.59 - 2.26) <0.001 1.93 (1.58 - 2.36) <0.001

Leukocytes – G/L 1.02 (0.97 - 0.98) 0.54 

Creatinine – μmol/L 1.00 (1.00 - 1.00) 0.09 

eGFR – mL/min 1.01 (1.00 - 1.01) 0.07 

BNP at baseline† – pg/mL 0.97 (0.95 - 0.99) <0.001 0.97 (0.95 - 0.99) <0.001

CT-ADP – sec 1.00 (1.00 - 1.00) 0.74 

PT – % 1.0 (1.00 - 1.00) 0.60 

LVEF – % 1.31 (0.58 - 2.95) 0.51 

Mean aortic gradient – mmHg 1.00 (0.99 - 1.01) 0.55 

Aortic valve area – cm2 1.23 (0.73 - 2.07) 0.43 

† per 100 pg/mL

Abbreviations: APT = antiplatelet therapy; BMI = body mass index; BNP = B-type natriuretic peptide; CI = confidence interval; CT-ADP = closure time adenosine diphosphate; eGFR = estimated glomerular filtration rate; LVEF = left ventricle ejection fraction; MPV = mean platelet volume; OR = odds ratio; PC = platelet count; PT = prothrombin time.

- Please verify the HR and 95%CI reported for platelets at baseline.

The HR and 95%CI of platelets at baseline are shown above (Tables 4 and 5). We now indicate the HR of platelet levels per 100 G/L.

- Have not normally distributed variables been ln-transformed before entering regression models?

Although we acknowledge that it could be one of the methods to analyze the relationship between non-distributed variables and endpoints, we preferred to keep the present form because it would be more reader friendly (e.g. the effect of 100 pg/ml increase of BNP) and would be consistent with our prior studies (e.g. Matsushita K, et al. Thrombosis and Haemostasis 2020). We appreciate your understanding.

- Units of measure should be reported for each variable in Table 4 and 5.

We added the units of measure for each variable concerned Tables 4 and 5.

- How were variables selected to entering the regression models?

As described in the “Methods” section of the main text (Page 10, Lines 18-19), we selected in the multivariable model variables with a p-value <0.05 in the univariate analysis.

---

## [Decision Letter · Decision Letter 1]

10 Nov 2021

Baseline Mean Platelet Volume is a strong predictor of major and life-threatening bleedings after Transcatheter Aortic Valve Replacement

PONE-D-21-27214R1

Dear Dr. Trimaille,

We’re pleased to inform you that your manuscript has been judged scientifically suitable for publication and will be formally accepted for publication once it meets all outstanding technical requirements.

Kind regards,

Alberto Aimo, MD

Academic Editor

PLOS ONE

Additional Editor Comments (optional):

Reviewers' comments:

Reviewer's Responses to Questions

**Comments to the Author**

1. If the authors have adequately addressed your comments raised in a previous round of review and you feel that this manuscript is now acceptable for publication, you may indicate that here to bypass the “Comments to the Author” section, enter your conflict of interest statement in the “Confidential to Editor” section, and submit your "Accept" recommendation.

Reviewer #2: All comments have been addressed

2. Is the manuscript technically sound, and do the data support the conclusions?

Reviewer #2: (No Response)

3. Has the statistical analysis been performed appropriately and rigorously? 

Reviewer #2: (No Response)

4. Have the authors made all data underlying the findings in their manuscript fully available?

Reviewer #2: (No Response)

5. Is the manuscript presented in an intelligible fashion and written in standard English?

Reviewer #2: (No Response)

6. Review Comments to the Author

Reviewer #2: (No Response)

7. PLOS authors have the option to publish the peer review history of their article (what does this mean?). If published, this will include your full peer review and any attached files.

Reviewer #2: No

---

## [Editor Report · Acceptance letter]

15 Nov 2021

PONE-D-21-27214R1 

Baseline Mean Platelet Volume is a strong predictor of major and life-threatening bleedings after Transcatheter Aortic Valve Replacement 

Dear Dr. Trimaille:

I'm pleased to inform you that your manuscript has been deemed suitable for publication in PLOS ONE. Congratulations! Your manuscript is now with our production department. 

Kind regards, 

on behalf of

Dr. Alberto Aimo 

Academic Editor

PLOS ONE